# Attitudes and Approaches to Use of Meal Replacement Products among Healthcare Professionals in Management of Excess Weight

**DOI:** 10.3390/bs10090136

**Published:** 2020-09-07

**Authors:** Gabrielle Maston, Janet Franklin, Alice A. Gibson, Elisa Manson, Samantha Hocking, Amanda Sainsbury, Tania P. Markovic

**Affiliations:** 1Faculty of Medicine and Health, The University of Sydney, The Boden Collaboration for Obesity, Nutrition, Exercise & Eating Disorders, Charles Perkins Centre, Camperdown, NSW 2050, Australia; samantha.hocking@sydney.edu.au (S.H.); tania.markovic@sydney.edu.au (T.P.M.); 2Metabolism & Obesity Services, Royal Prince Alfred Hospital, Camperdown, NSW 2050, Australia; janet.franklin@health.nsw.gov.au (J.F.); elisia.manson@health.nsw.gov.au (E.M.); 3School of Public Health, Faculty of Medicine and Health, Centre for Health Policy, The University of Sydney, NSW 2050, Australia; alice.gibson@sydney.edu.au; 4Faculty of Science, School of Human Sciences, The University of Western Australia, Crawley, WA 6009, Australia; amanda.salis@uwa.edu.au

**Keywords:** obesity, diet, reducing, weight management, low-energy liquid diet, qualitative, very-low-energy diet

## Abstract

Meal replacement product-based diets are an effective weight loss intervention used in the management of obesity. Historically, these diets have been underutilised by HealthCare Professionals (HCPs). An online survey of mixed methods design was distributed to HCPs to capture current perceptions and prescribing patterns of meal replacement products (MRPs) in the management of overweight and obesity. A total of 303 HCPs working in weight management across Australia began the survey and 197 (65%) completed it. While over 70% of HCPs have prescribed MRP currently or in the past, MRPs are only prescribed to a median 7% of patients seeking weight management treatment. Qualitative analysis identified potential barriers to MRP prescription, which include experience with patient non-compliance, perceived poor long-term weight loss durability and safety concerns regarding the product and its use as a total meal replacement program. Safety concerns are centred on the perceived risk of weight cycling and its potential negative psychological impact. MRP prescription is 66% more likely to occur if HCPs had formal training in the use of MRPs relative to those who did not, with a relative risk (RR) of 1.7 (95% CI 1.4, 2.0). This study highlights the potential barriers to the prescription of MRPs, which are centred around safety concerns. This also indicates that formal training may enhance the likelihood of prescribing MRPs, suggesting that once HCPs have a comprehensive understanding of the products and the evidence behind their use, their prescription is likely to be increased.

## 1. Introduction

Formulated Meal Replacement Products (MRPs) have been used extensively in the management of overweight and obesity (body mass index [BMI] ≥ 25 kg/m^2^) for more than 40 years [1]. Historically, formula MRPs have been prescribed in the form of a very-low-energy diet (VLED). VLEDs are commonly implemented as a meal replacement diet, involving the exclusive use of MRPs and sometimes additional low-energy containing food items, such as low-starch vegetables, black tea and coffee [2]. VLEDs are defined internationally by the World Health Organisation, international CODEX standardization by the United States of America (USA) Food and Drug Administration and the European Union as formula-based foods used in weight control diets that provide between 2090 to 3350 kJ (500 to 800 kcal) daily [3,4,5]. MRPs can also be used during a low-energy diet (4200 to 5000 kJ, [1000 to 1200 kcal]), commonly prescribed as a combination of MRPs and energy-controlled food-based meals [6].

The inception of VLEDs stemmed from an interest in creating a dietary weight loss therapy that could produce outcomes comparable to starvation but without the increased risk of ill health or death [1]. The precursor of MRPs began in 1915 when high protein food-based diets of 1700 to 2500 kJ (400 to 600 kcal) were used for rapid weight loss to treat obesity [7]. Liquid drinks comprised almost entirely of high-quality protein (casein or egg albumin) or a combination of protein and carbohydrate (egg albumin and sucrose or glucose) were later developed to create a semi-starvation state with success [1,8,9]. However, in the late 1970s, liquid milk formula-based diets began to be produced from low-quality protein (hydrolysed collagen or gelatine) that was nutrient incomplete [8] and led to 60 deaths in the U.S.A [8,10]. The deaths were later attributed to cardiac complications resulting from a starvation response from prolonged and extremely rapid weight loss (~30% body weight in 4 months) in otherwise healthy young people with obesity [10,11].

High-quality protein and nutrient replete MRPs are now used [7,12] and clinical trials from the early 1980s and ongoing, show that VLEDs using MRPs are a safe and highly effective weight loss intervention for the management of obesity [1,12,13,14,15].

Despite the redevelopment of MRPs, the adverse historical events seemed to shape the perception of such diets among HealthCare Professionals (HCPs) into the 21st century [7,16]. In the past decade, three separate surveys exploring general weight management techniques used by HCPs, show that MRPs and VLEDs remain underused. Only 3.2% of dietitians were found to prescribe VLEDs as a weight loss therapy [16]. Similarly, just 2% of surveyed British physicians reported they would advise patients to use a VLED [17] and among approximately 1200 surveyed physicians in the USA, meal replacement diets were the least used weight loss therapy [18].

The underutilisation of MRPs is surprising due to the success of meal replacements diets reported in several clinical trials published during the time these surveys were conducted (1991 to 2003) [13,19,20,21,22,23,24,25]. On average VLEDs produce 2 kg weight loss per week acutely in approximately 80% of individuals [1,13] and are a more effective long-term mode of weight loss than a food-based energy-reduced intervention [26]. There is a 3.1-fold greater likelihood of losing 10% clinically significant weight loss at 24 months after a four month meal replacement diet of 65 to 75% energy restriction (followed by moderate energy restriction for 8 months), compared to a 12 month 25–35% moderate energy-reduced food-based diet [26]. The success of meal replacement diets in producing safe [27] and predictable weight loss [10,13] has led to their widespread commercialisation [28]. The wide commercialisation and availability of MRPs indicate that there is prevalent use among the general public, demonstrating its acceptability as a weight loss solution, yet this does not appear to be so among HCPs.

Poor acceptability of MRPs by HCPs may be due to conflicting clinical practice guidelines around the world [2,29]. Australian clinical practice guidelines for the management of overweight and obesity state that VLEDs may be a suitable option for people with overweight and obesity (BMI > 27 kg/m^2^ with obesity related co-morbidities) when used in a medically supervised weight loss program [6]. In contrast, such diets are not recommended in the routine management of obesity in UK NICE guidelines, European Guidelines for Obesity Management nor in the American College of Cardiology Task Force guidelines in the USA [4,30,31]. HCPs who follow clinical practice guidelines are thus unlikely to prescribe MRPs. However, recent evidence shows that meal replacement diets are effective [15] and therefore should be considered for weight management.

Whist the evidence is clear that meal replacement diets do assist with short and long-term weight loss, there is also conflicting evidence to show that at least 56% of weight lost during a meal replacement diet is regained after 12 months [32] which may also be influencing their use amongst HCPs.

The attitudes and prescription patterns surrounding MRPs have not been formally explored in more recent years. This project aimed to explore attitudes and approaches to the use of MRPs among HCPs with quantitative and qualitative methods to identify potential factors that may hinder and facilitate their use.

## 2. Methods

### 2.1. Study Design and Participant Recruitment

Ethics was approved by the Ethics Review Committee Royal Prince Alfred Hospital of the Sydney Local Health District on 7 September 2017. LNR-NSW (AU/6/8BFE23). All participants were provided with information and gave consent prior to commencing the survey.

A cross-sectional survey of Australian HCPs was undertaken between July and October 2018. An electronic invitation to participate in the survey was directed to HCPs who work in weight management using Research Electronic Data Capture (REDCap^®^). A targeted approach was used during our recruitment strategy to identify HCPs who worked in weight management.

HCPs were defined as university-trained medical professionals who worked in weight management be it in private practice, the public healthcare system, commercial business or research that provided a medical service in conjunction with weight loss advice.

Other HCPs without an expertise in nutrition were purposely targeted in our survey distribution strategy. Dietitians and nutritionists are often seen to be gate keepers of MRP prescription, however HCPs involved in multi-disciplinary teams do prescribe such diets and have some experience with patient compliance, behaviour and the satisfaction of meal replacement diets. This patient feedback relayed through other HCPs, outside of the dietetic practice, is valuable in providing a more robust understanding of prescription barriers.

A variety of methods was used for survey distribution and included providing an online link to professional medical associations, private businesses with professional membership, private businesses specialising in obesity management, personal and extended contacts and via social media platforms, including closed groups on Facebook and Twitter and direct emails to individuals who work in weight management. Private businesses with professional memberships included professional education providers, such as Dietitian Connection (10,000 members) and MegaBite Nutrition (membership base unknown). Professional medical associations included the Dietitians Association of Australia Obesity Interest Group (membership base unknown), Exercise Sports Science of Australia private Facebook groups (1200 members), Network of emerging Australian dietitians Facebook group (2600 members), The Australian Primary Care Nurses Association newsletter (membership base 66,000; newsletter distribution list unknown), and newsletter correspondence to all Primary Health Networks throughout Australia (membership base unknown). Extended personal contacts included HCPs from obesity weight management outpatient services in public hospitals across Australia and HCPs in private practice businesses.

The survey was also produced in a hard copy format and used for opportunistic sampling by approaching HCPs attending obesity-related professional development workshops. Responders were encouraged to share the survey with colleagues working in weight management resulting in recruitment by snowballing [33].

During the dissemination period, repeated contact was made to optimise the number of respondents. This included three reminder emails to points of contact and repeated exposure to marketing material in online newsletters and on social media platforms. A response rate could not be calculated due to the inability to obtain private information pertaining to the size of the distribution lists of the private and commercial businesses or professional memberships and newsletter distribution lists.

On a weekly basis, data was monitored and preliminary coding applied to the qualitative responses to monitor for saturation. The open-text survey responses were monitored and analysed weekly for themes and repetition thereof. Although data saturation occurred when the 249th survey participant completed the survey, the survey was not discontinued due to the collection of quantitative data points contained in the survey. Data collection continued until we exhausted all recruitment avenues. This continued for four months. No personal identifying information was captured to ensure respondents could remain anonymous.

### 2.2. Measures

The survey titled “Attitudes and approaches to the use of meal replacement products amongst healthcare professionals in the management of excess weight” used a mixed-methods design. It consisted of both quantitative and qualitative questions to capture perceptions and experiences of HCPs in prescribing MRPs. The survey questions were developed by the authors and were underpinned by a literature review of clinical trials involving meal replacement programs and using prior knowledge from clinical practice [2,34]. Survey questions were peer-reviewed by ten external HCPs with expertise in weight management in two stages. During the initial stage, all suggestions were collated and incorporated into the survey. The survey was then sent to all reviewers for the final review, after which, further adjustments were made. The final survey contained 25 questions consisting of 10 multiple choice questions where participants could choose from one or more options, three dichotomous (yes/no) questions and 12 open-ended questions containing text fields with no limitation on text length. The three sections of the survey included demographic information and area of clinical practice (4 questions), prescription patterns (15 questions), and perceptions surrounding compliance, durability and safety of MRPs (five questions). The survey took, on average, 6 min (approximate range 1 to 20 min) to complete. A copy of the survey questions is provided as Appendix A.

### 2.3. Data Analysis

All statistical analyses were conducted using STATA/IC version 14·2, Windows 64 bit (StataCorp LP, College Station, TX, USA). All data available, including from incomplete surveys, were used in the data analysis. Descriptive statistics were determined for quantitative data and a hierarchical multiple regression analysis was used to identify factors that may predict MRP use. A correlation coefficient (*r*) was obtained that explained the strength of the relationship between MRP use and identified independent variable categories. This statistical model was conducted a second time transforming *r* into a standardised regression coefficient β. This allowed the relative strength of the identified independent variables to be compared to each other within the model. An adjusted R^2^ statistic (aR^2^) was calculated to determine how well the regression model explained the predictive power of the all the independent variables used the model.

Open-ended free-text responses were qualitatively analysed and categorised with use of NVivo v.12 software. An inductive approach was applied to create descriptive coding lists for each open-text response and secondary analysis was used across the data set to capture recurring broader themes [35]. An inductive approach was applied to create descriptive manual coding lists for each open-text response by lead researcher GM. When the inferred meaning of the open-text response could not be adequately determined, a second investigator, (J.F.), was consulted and after review, a code was applied or a new code was created.

The initial analysis included familiarisation of the data in open-text responses in the survey by reviewing each response, then reviewing a second time to create a list of descriptive categories for each question. In a third review, a code was then applied to each unique descriptive category and revised, and then descriptive categories were collapsed into broader theme-based categories. Responses were reviewed a fourth time to determine the fit of the responses to each code. Each collapsed category was then numerically quantified by the number of respondents and frequency in which the category was identified providing descriptive statistics to open-text survey responses. Further analysis included comparing theme-based categories across the entire survey questionnaire to further collapse categories into key overarching concepts. This process avoided the repetition of results and allowed for the identification of potential barriers to the use of MRPs among HCPs and how the prescription of MRPs differed from prescription patterns observed in clinical trials.

The survey was designed by investigators G.M., J.F. and A.A.G., who are experienced clinical dietitians with limited experience in qualitative methodology. G.M. is completing her doctoral degree in research and J.F. and A.A.G. are experienced post-doctoral researchers. To mitigate researcher bias, the following methods were used: J.F. reviewed the code lists and sections of raw data when discrepancies arose. This process was used to determine consistency between the primary researchers’ interpretation and the interpretation from the second data reviewer. During the survey design process, analysis of data and when presenting proposed research outcomes, supervision was used. Associate professors A.S., T.M. and S.H. provided supervision and have extensive research experience and expertise in the area of obesity management and MRPs. The supervision and consultation processes were implemented for the design, implementation, collation of data and formulating arguments. During the consultation process, gaps were identified by each of the three supervisors separately, implemented and reviewed until no more discrepancies could be found.

## 3. Results

### 3.1. Participant Characteristics and Demographic Information

A total of 303 HCPs began the survey and 197 (65%) completed it. The majority of responders were captured via REDCap^®^ (89%), with the remaining 11% of responders from in-person paper-based surveys. All responses were included in our data analysis. There were variations in the number of questions answered in each survey as HCPs were able to skip questions or abandon the survey as desired. A response rate could not be determined due to the online recruitment strategy.

A wide range of health professionals completed the survey. Distribution of HCP occupation, employment location area and type of employment setting is shown in Table 1. Occupation was provided by 269 HCPs.

HCPs were retrospectively grouped into the following categories; (i) Dietitians and nutritionists, (ii) Allied health professional (excluding dietitians): physiotherapists, exercise physiologists and exercise scientists, psychologists and counsellors, sonographers, diabetes educators, (iii) Medical: general practice physicians and nurses, (iv) Specialist medical: endocrinologists, gynaecologists, gastroenterologists, bariatric endoscopists and paediatricians, (v) Other HCPs: researchers, pathology laboratory technicians, lymphoedema therapists and public health practitioners. The majority, 78% (*n* = 211), of the HCPs surveyed were dietitians and/or nutritionists. There were 45 HCP who identified with more than one profession. The type of employment setting was provided by 264 participants, with the majority (51%, *n* = 135) of HCPs working in private practice and 120 identifying with more than one employment setting.

### 3.2. Prescription Patterns

The most common responses are highlighted below. The full results of the survey for prescription patterns are provided in Table 2.

#### 3.2.1. Have You Ever Used Meal Replacement Products to Help Your Patients/Clients Manage Excess Weight Currently or in Past?

The majority (73%, *n* = 175 of 241) reported prescribing MRPs. For those who did not use MRPs, the main reasons provided were: (i) 46% (28 of 61) felt they lacked knowledge in product and program safety with dietary prescription falling outside their scope of practice; (ii) 23% [14] preferred to use lifestyle behaviour change as weight management therapy; (iii) 12% [7] considered MRPs an unsustainable long-term solution to weight management.


*“I am not confident/trained in their use or in prescribing a full nutritional diet. So, I would always aim to work in with an expert in diets for this reason. Till I expand my knowledge base”. *
*(General practitioner)*


*“I’m a strong believer in holistic lifestyle change. I work with patients with chronic diseases and I believe that meal replacements, although I’m sure have their place in high risk and low weight, are not well utilised in managing overweight or obesity. Many patients make poor food choices chronically. At times with these persons, meal replacement works temporarily but as soon as they are finished the intervention they return to unhealthy behaviours and weight gain returns. Often psychological support and proper nutritional planning is required for long term health changes”. *
*(Dietitian)*


*“Not a holistic approach, ignores the reasons people are eating in the first place and when it comes down to it, meal replacements are not ‘real’ (‘whole’) food or sustainable in the long run” *
*(No profession provided)*


*“I think they are pretty terrible actually. I can get the idea of a modified fast for short-term loss, but it doesn’t teach the patient anything about eating and conflicts with all our other messages about food”. *
*(Dietitian)*

#### 3.2.2. What Percentage of Your Total Patient Load Requiring Weight Loss Are Prescribed a Diet Containing Meal Replacement Products?

A total of 233 responses were received, ranging from 20% (47) HCPs who did not prescribe MRPs at all to 4% (9) HCPs who prescribed them to all patients seeking weight-loss treatment. MRPs were prescribed by a median of 7% (mean 19%) of the HCPs’ total patient load seeking weight-loss.

#### 3.2.3. Have You Ever Had Formal Training on How to Prescribe and Use Meal Replacement Products as a Strategy for Weight Management?

There were 58% (140) HCPs who had formal training on how to prescribe MRPs. MRP prescription increased by 66% (RR 1.7, 95% confidence interval [CI] 1.4, 2.0) when HCPs had formal training, compared to those that had not. The types of formal training commonly undertaken were lectures at university (56%, *n* = 76) and conferences/workshops/webinars (42%, *n* = 57).

A hierarchical multiple regression analysis was used to determine if there were variables, along with formal training, that could predict MRPs prescription. The following variables were significantly correlated with MRPs prescription (*p* < 0.05) in order of predictive strength: (i) HCP occupation: dietitian/nutritionist (β = 0.41) and endocrinologist (β = 0.22); (ii) formal training (β = 0.20); (iii) working in private practice (β = 0.19); (iv) working in an outpatient hospital environment (β = 0.16). These variables accounted for 32% of the variability in MRP prescription (aR^2^ = 0.3).

#### 3.2.4. What Factors Determine What Type of Patients/Clients You Prescribe Meal Replacement Products for?

HCPs were provided a multi-select multiple choice question that included a list of factors that may determine MRP prescription. The two most common factors reported by the 239 respondents were "weight loss required prior to surgery" (the type of surgery was not asked) reported by 67% (160) and the severity of obesity for 66% (158).

#### 3.2.5. Have You Ever Recommended Clients Undergo a Rapid Phase of a Meal Replacement Diet to Lose Weight?

The rapid phase of a meal replacement diet was defined in the survey as: a period which consists of the sole consumption of MRPs (often referred to as the intensive phase). There were 217 responses to the question, 43% (94) responded positively to prescribing the rapid phase of a meal replacement diet to their patients.

#### 3.2.6. How Do You Typically Prescribe Meal Replacement Products during the Rapid Phase of a Meal Replacement Weight Loss Program?

A response where two main themes emerged was provided by 79% (*n* = 74 of 95)—49% (36) followed Optifast^®^ intensive level guidelines (Nestlé Health Science) and 18% (13) prescribed additional protein to meet a predetermined protein target. The prescription of additional protein was broken down into three different categories: (i) three MRPs with additional protein at the start of the meal replacement diet (1%, *n* = 1); (ii) three MRPs with additional protein throughout the entire meal replacement diet (8%, *n* = 6); (iii) the number of MRPs prescribed was adjusted to meet individual protein requirements (8%, *n* = 6).


*“Lengthy discussion re [regarding] protocol of following VLED diet. Intensive phase (i.e., typically 3 meal replacements [per] day, more if someone with a higher muscle mass) used for rapid phase. Patient handout sheets provided”. *
*(Dietitian and Diabetes educator)*

#### 3.2.7. How Many Meal Replacement Products Do You Typically Prescribe as a Daily Intake When the Goal is Rapid Weight Reduction?

There were 217 responses, of which 54% (76) prescribed ≤ 3 meal replacement products per day and 33% (47) varied the prescription according to the individual’s height and/or weight.

HCPs who adjusted their dietary prescription to the individual’s anthropometry were provided with an open-ended question and asked to describe their method of prescription. The most common response was the adjustment of prescription according to the individual’s protein requirements (57%, 25), and this was done using an adjusted ideal body weight by 16% (*n* = 7) of respondents.


*“Calculate energy requirements for weight, height and activity level, and required energy deficit for weight loss. Calculate how many meal replacement products are needed to meet this energy requirement”. *
*(Dietitian)*


*I determine the number of MR [meal replacements] required based on protein requirements and discuss with the client the number of products vs. low carb meals they would like in the plan (usually for those requiring more than 3MR [3 meal replacements])”.*
*(Dietitian)*

#### 3.2.8. Do You Ever Allow Additional Items in a Meal Replacement Diet as Part of Your Prescription for Weight Loss?

Ninety eight percent (212) of respondents prescribed additional items during a meal replacement diet—only 2% (5) did not. The two most commonly prescribed food items were (i) non-starchy vegetables (e.g., spinach, broccoli, tomato, salad or vegetable soup) (67% *n* = 146) and (ii) food-based protein (e.g., meat, fish, eggs, chicken, pork or tofu) (44%, *n* = 95).

### 3.3. Perceptions of Compliance, Durability of Weight Loss and Safety

The most common responses to the section, perceptions of compliance, durability of weight loss and safety, are highlighted below. The full results of the survey for this section are provided in Table 3.

#### 3.3.1. Have You Ever Experienced Patient Non-Compliance with Diets Involving Meal Replacement Products?

HCPs’ experience with non-compliance was high with 72 (83%) reporting they had experienced patient non-compliance.

#### 3.3.2. What Reasons Do You Believe Contribute to Patient Non-Compliance?

HCPs believed that following a highly restrictive diet that limited carbohydrate and/or energy intake was extremely difficult to maintain, with (9 of 63) 14% stating the diet was too restrictive. The difficulty was believed to be due to boredom and emotional eating 19% (12), patient food taste and texture preferences (e.g., preferring to consume chewable and palatable food, 16% (10), the difficulty in implementing and maintaining behaviour change, 14% (9), hunger 11% (7), and the ongoing cost of purchasing meal replacement products, 11% (7).


*“Boredom, not addressing reasons for overeating—habit/emotional reasons etc. leading to binge, social events” *
*(Diabetes educator and Dietitian)*


*“Patients miss being able to chew their food, don’t like the taste/texture of the meal replacement, the meal replacements don’t keep patients satiated” *
*(Dietitian)*


*“It’s a big transition from eating large quantities of food to eating little. Food boredom, social preferences, emotional eating and cost. It’s a very difficult diet to be compliant with i.e., socially isolating, not flexible. Also, w/[with] clients prescribed this diet, they often have been previously non-compliant and/or have limited social support available at home” *
*(Dietitian and Exercise Physiologist)*

#### 3.3.3. What Is Your Experience with Patients Regarding the Long-Term Outcome of Weight Loss and Weight Maintenance When It Is Achieved with a Meal Replacement Diet?

Forty nine percent (99 of 202) of responders reported weight loss achieved during a meal replacement diet was not durable and 23% (46) reported weight loss durability was conditional. The main reasons provided by 48% (47 of 99) for perceived poor long-term weight loss durability were (i) meal replacement diets do not encourage permanent behaviour change (53%, *n* = 25); (ii) the diet was highly restrictive diet and thus unsustainable (26% *n* = 12); (iii) that all restrictive weight-loss diets are ineffective and weight regain is inevitable (13%, *n* = 6).

In the subgroup of HCPs who reported that weight loss durability was conditional, responses included multiple overlapping factors. These factors included achieving the desired weight loss goal, an improvement in lifestyle behaviours, access to ongoing support and education from a HCP, access to appetite suppressant medication, and following the appropriate transition phases of moving a person from consuming MRPs to eating whole food.


*“In the long term, relapse often occurs. The pt [patient] achieves weight loss but is not able to maintain it for very long. In general, some people regain weight as the behaviours that contributed to the initial weight issues were not resolved”. *
*(Dietitian)*


*“Long-term outcomes appear to better if issues/barriers around nourishing eating patterns/styles can be addressed during a VLED as we move into ‘normalising’ the program e.g., emotional eating, hectic schedules, poor planning etc. My biggest concerns around ‘dieting’ with the use of meal replacements & in general is possible re-enforcement of dichotomous thinking around food, calorie counting etc. I try to engage with clients & their thoughts/behaviours around a healthy relationship with food”. *
*(Dietitian)*


*“Unless a dramatic improvement (as defined by the patient) is achieved or they are used chronically in weight management there is commonly rebound gain” *
*(Nurse)*


*“Must include slow transition back to normal meals, very slow re-intro [introduction] CHO [carbohydrates], +/− [with or without] medication to assist appetite suppression” *
*(Nutritionist)*

#### 3.3.4. What Are Your Perceptions about the Safety of Meal Replacement Programs as a Weight-Loss Tool?

Of the 196 responses, 37% (72) thought they were safe, 31% (60) believed they were safe with medical supervision and 21% (41) considered safety conditional on other factors. The conditional factors included the individual’s ability to adopt healthy eating behaviours when MRP use ended, that they were only prescribed to people with low-risk medical conditions or no co-morbidities and conducted with a particular product brand. While acknowledging that MRPs were physiologically safe, there was the belief that they could cause psychological harm because of the restrictive nature of meal replacement diets and the resulting weight cycling that may occur. Of the remaining HCPs (12%, *n* = 23), who believed meal replacement diets were not safe, there were no commonalities between responses. Singular responders questioned the safety of the products concerning electrolyte imbalances, muscle wasting, body dissatisfaction, and the formation of gallstones. Others believed the restrictive formula-based diet conflicted with positive messages about whole food stated in population guidelines for healthy eating.


*“As long as a reputable brand is used and under supervision of a dietitian and guidelines adhered to I would believe they are safe” *
*(General practitioner)*


*“They can be effective with the correct support and education around using them appropriately. It depends on the individual for the safety for the meal replacement as they are not suitable for everyone for example, adolescents or various chronic diseases. Using meal replacements require professional judgements to assess the safety of use for each individual and tailor meal plan or examples of use as required”. *
*(Dietitian and Nutritionist)*


*“If individual progress is monitored well, and the program incorporates healthy eating I believe they have a place as a weight-loss tool, however, this isn’t the case in a private setting and often unqualified people are promoting them to the public. If doctors prescribe meal replacement shakes they should also be given automatic subsidised visits to a dietitian to ensure safety”. *
*(Dietitian and Nutritionist)*


*“I believe meal replacements are technically safe (won’t kill the patient) but I don’t believe meal replacements are a viable solution or supportive for health. In fact, I feel the use of meal replacements and the psychological impact caused to the relationship with food is harmful. As I do not wish to do harm to my patients, I would not prescribe meal replacements”. *
*(Dietitian)*

## 4. Discussion

Our results demonstrate that 73% of HCPs had experience in the use of MRPs, however MRP prescription was only used for a median of 7% of all patients seen for weight management. Thus, it appears that the prescription of MRPs is not the main weight loss therapy used. The low rate of prescription seems to be influenced by the attitudes held by HCPs about MRPs and included concerns that they attract a higher level of non-compliance because of their restrictive nature, they have poor long-term weight loss outcomes and that they are not safe because of the risk of weight cycling and a negative psychological impact. These results and previous investigations suggest there is a disconnect between the evidence base and HCPs’ attitudes and approaches to clinical practice in the use of MRPs for weight management [36].

Eighty three percent of HCPs have had experience with poor patient dietary compliance to a prescribed meal replacement diet, and this is likely to negatively influence their prescription of MRPs. While compliance to all dietary weight-loss interventions diminish over time [37], in clinical trials meal replacement diets have greater compliance and produce greater weight loss at 12 months when compared to food-based dietary interventions [38,39]. It appears this finding is not well understood by the broader HCP community. It may also be a reflection of the general messages stated in population guidelines for healthy eating [40], rather than clinical practice guidelines for the management of overweight and obesity [6]. Our results found that clinical practice guidelines are the least used mode of training by HCPs to inform the use of MRPs, thus it is unlikely that conflicting international clinical practice guidelines are the reason for MRP underutilisation. An improvement in dissemination of the Australian clinical guidelines may enhance MRP prescription amongst Australian HCPs. HCP experience with poor compliance may also be influenced by negativity bias. Negativity bias is a basic tenet of psychology, in which negative information outweighs positive information, thereby having a greater effect on impression as it is more likely to be recalled by memory [41,42]. HCP experiences with poor dietary compliance may be influenced by a degree of perceptual bias.

Another barrier to MRP prescription was the perception of poor long-term weight loss maintenance, with 49% of HCPs reporting that weight loss achieved during a meal replacement diet was not durable. In comparison to hypo-energetic food-based dietary interventions, severely energy-restricted meal replacement diets are more effective at weight loss and weight loss maintenance [26,43]. One long-term clinical trial demonstrated that at 3 years, after 4 months of severely energy-restricted meal replacement diet, 7.3 kg weight loss was maintained long-term. In contrast, only 3.2 kg weight loss was maintained after a 12 month hypo-energetic food-based intervention at the 3 year time point [26].

The disconnect between the evidence base and perceptions among HCPs may be due to the conflicting weight loss outcomes of older publications that have reported weight loss is no greater than food-based interventions long-term [44] and that weight rebound can occur after meal replacement diet interventions [45] as well as more recent papers reporting the ineffectiveness of self-directed commercial meal replacement diets without HCP support [46]. Until recently, the use of adjunctive therapies, such as behavioural lifestyle therapy and pharmacotherapy, were less commonly used in meal replacement diet programs [2]. In addition, self-directed dieting for weight loss has been shown to be less effective long-term than HCP supported dieting [47]. Participants volunteering for clinical trials also tend to be highly motivated to achieve health outcomes and may be incentivised by free healthcare or monitory gifts [48,49,50]. Therefore, the results achieved in clinical trials may not reflect those achieved by people seen in real-world clinical settings, and this is a key issue that requires investigation.

MRP safety has historically been an area of concern for HCPs [7,28]. While our survey demonstrated that safety concerns remain, the nature of these has changed and now relates to the restrictive nature of the diet and the potential for weight cycling rather than direct toxicity from the MRPs. Weight cycling, which is repeated bouts of intentional weight loss followed by weight regain, has not been shown to cause measurable psychological harm in individuals with obesity who seek weight loss treatment—rather weight cycling is thought to be a result of pre-existing psychological factors [51,52,53,54]. Contrary to the popular perception among HCPs, weight regain is not related to the severity of dietary energy restriction [26]. Formal education and training may be a way to improve this misconception among HCPs. We have demonstrated that the use of MRPs was 66% more likely when HCPs had undergone formal training. Therefore, providing professional education courses, workshops or conferences on MRP use and prescription may be an avenue to improve attitudes and approaches towards such products. The regression analysis also demonstrated that dietitians and endocrinologists are more likely to use MRPs than other HCPs and this may be because these professions are more likely to be trained in this area. Formal training may be the most important factor leading to MRP use.

This survey demonstrated that the majority of HCPs prescribe MRPs according to prescription patterns observed in clinical trials, which consists of three MRPs per day with or without the use of supplemental low energy food, such as low starch vegetables [2,28]. A minority (18%) of HCPs prescribe MRPs with additional protein supplementation and/or protein-rich food-based items during the rapid phase. Higher protein intakes (≥1 g protein per kilogram body weight) are thought to assist in appetite control [55] and the maintenance of fat-free mass (FFM) during intentional weight loss [56,57]. However, the evidence to support the addition of protein to a meal replacement diet is conflicting [14,58] and limited, and thus warrants further investigation.

There are several limitations to be considered when interpreting the results of this survey. Primarily, the number of responders beyond the professional category Dietitian and Nutritionist was low, and this affects the generalisability of results to other professions. The low representation (under 40%) from HCPs outside of the dietetic and nutrition profession may be because of the lack of relevance of the survey to HCPs without an interest in nutrition and may also reflect their lack of familiarisation with MRPs. However, the under-representation of other HCPs is not surprising as dietitians and nutritionists usually conduct dietary interventions in the Australian healthcare system, with referrals from other allied health and medical professionals. As seen by several of the comments in this survey, the prescription of MRPs is viewed as a speciality area of dietitians and nutritionists within a multidisciplinary team.

There is potential for bias in the survey responses due to the language used to describe meal replacement diets in the survey. The survey instrument used terms such as “meal replacements” and “rapid phase”, which are generic terms used to describe such programs compared to terms such as “VLED” and “intensive phase”, terms more commonly used to describe such diets. Our approach was taken to capture the wide variation of MRP use and to reduce bias towards particular branded products and commercial protocols. These slight differences in vocabulary may have solicited variations in responses that may affect the interpretation of our findings. The survey instrument was not formally tested for reliability or validity, increasing the risk of measurement error and random response error. An effort was made to carefully construct survey questions to minimise error through a peer review process in the development of the survey questions and answers selection, but we could not control for individual differences in the interpretation of questions.

Another limitation identified was the potential for bias resulting from the online snowballing recruitment method. There is potential that HCPs from the same workplace, and thus with the same workplace practices, responded to the survey. In allowing respondents to remain anonymous, the proportion of participants who responded from the same work place could not be quantified, thus recruitment bias may have influenced the results. There is also the potential for the survey to be initiated and completed by different HCPs within the one survey, thus reducing the accuracy of our results. In addition, we did not capture information pertaining to participant personal characteristics, such as employment duration, sex and age, which may also shape attitudes and approaches to the use of MRPs. Lastly, we acknowledge that, although an attempt was made to capture rich qualitative data, responses to the open text survey questions were short and brief in nature, which may reduce the generalisability of our results.

In contrast, we are confident that we obtained a wide variety of HCP perceptions and a large sample size indicated by reaching a saturation of answers in the open-text field responses. Lastly, we captured diversity in HCPs across Australia in both urban and remote geographical locations, as well as a variety of workplace settings.

## 5. Conclusions

While over 70% of HCPs surveyed have prescribed MRPs currently or in the past, they are only prescribed to a median of 7% of patients seeking weight management treatment. Barriers to MRP prescription include experience with poor dietary compliance, the perceived inability to sustain long-term weight loss and MRP program safety, most notably a perceived negative psychological impact of weight cycling and long-term weight regain. Prescription patterns are mostly congruent with those observed in clinical trials (three MRPs daily to replace each main meal), with the addition of low starch vegetables and, to a lesser extent, additional protein through supplementation or protein-rich food. Formal training is positively associated with MRP prescription and may enhance the likelihood of prescribing MRPs, suggesting that once HCPs have a comprehensive understanding of the products and the evidence behind their use, their prescription is likely to be increased.

## Figures and Tables

**Table 1 behavsci-10-00136-t001:** Survey participant characteristics.

**Distribution of Professional Occupations**	***n***	**%**
Dietitian and nutritionist	211	78
Allied health	46	17
Medical	29	11
Specialist medical	11	4
Other healthcare professionals	20	7
Total	269	(a)
**Distribution of Participants across AUSTRALIAN States**	***n***	**%**
Victoria	10	3
Northern Territory	5	2
Queensland	11	4
South Australia	3	1
Western Australia	13	5
Tasmania	42	14
Australian Capital Territory	44	15
New South Wales	137	45
Total	265	(b)
**Type of Practice Location Area**	***n***	**%**
City	112	37
Urban area	97	32
Regional	53	18
Remote	8	3
Total	264	(b)
**Type of Employment Setting**	***n***	**%**
Private practice/rooms	135	51
Hospital outpatient	74	28
Hospital inpatient	73	28
Community health	56	21
Telecommunication health	16	6
Gymnasiums	8	3
Academics/researchers/students	22	8
Total	384	(a)

Abbreviations: *n*, number; %, percentage. Values marked (a) are greater than 100% as 45 healthcare professionals had more than one professional title or worked in more than one employment setting. Values (b) do not add up to 100% due to 38 healthcare professionals’ non-disclosure of this information.

**Table 2 behavsci-10-00136-t002:** Healthcare professionals’ experiences and uses of meal replacement products in daily practice.

**Have you ever used meal replacement products to help your patients/clients manage excess weight currently or in the past?**			
	*n*	Yes	%
Total	241	175	73
**What are the reasons you have not used meal replacement products as a weight management strategy for patients/clients with excess weight? *(open text)***			
	*n*		%
Lacked knowledge in product and program safety with dietary prescription falling outside of the scope of practice	28		46
Preference for promoting lifestyle behaviour change	14		23
MRPs are an unsustainable long-term solution to weight management	7		12
Preference for prescribing "real whole food" rather than a formulated powder	6		10
Eating behaviours are attributed to a person’s psychological relationship with food, MRPs are an inappropriate solution to address this psychological issue	4		7
Preference to practice from a non-dieting weight neutral service philosophy	2		3
Total	61		
**Have you ever had formal training on how to prescribe and use meal replacement products as a strategy for patients/clients with excess weight?**			
	*n*	Yes	%
Total	241	140	58
Describe what type of formal training you have undertaken? *(open text)*			
	*n*		%
Lectures at university	76		56
Conference/workshop/webinar	57		42
Optifast^®^ accreditation course	23		17
Optifast^®^ commercial promotional material	11		8
On the job experience	9		7
Read clinical practice guidelines	1		1
Total	177		
**What factors determine what type of patients/clients you prescribe meal replacement products for?**			
	*n*		%
Weight-loss required for surgery (the type of surgery not specified)	160		67
Severity of obesity	158		66
Other obesity-related comorbidities	90		38
Type 2 diabetes	86		36
Level of support available	72		30
Fatty liver disease	62		26
Waist circumference	60		25
Age	57		24
Metabolic disease	53		22
Medications	51		21
Other *	45		19
Breathing difficulties	36		15
Gender	15		6
*** Specify what "other" factors determine what type of patient/clients you prescribe meal replacement products for? *(open text)***			
	*n*		%
After multiple failed weight loss attempts	12		30
Indicated by referring doctor or surgeon	9		23
At the patients’ request	4		10
If the patient’s lifestyle does not accommodate for cooking and meal preparation	4		10
In preparation for bariatric surgery	3		8
The need for pain management	2		5
Finances	2		5
If a patient has poor nutrition knowledge and skills	2		5
Infertility	2		5
Lack of weight loss motivation	1		3
Renal disease	1		3
Disordered eating patterns	1		3
Total	40		
**Have you ever recommended clients undergo a rapid phase of a meal replacement diet to lose weight? (Excluding before bariatric surgery)**			
	*n*	Yes	%
Total	217	94	43
**How do you typically prescribe meal replacement products during the rapid phase of a meal replacement weight loss program? *(open text)***			
	*n*		%
I follow the Optifast^®^ protocol that includes the use of three MRPs, low starch vegetables and one tsp oil daily	36		49
Additional protein is prescribed with three meal replacement products	13		18
Prescription is in negotiation with patient	3		4
Prescription varies for every person (more than three shakes and extra food)	12		16
Two shakes and one meal is prescribed to everyone	9		12
The number of meal replacements is based on achieving a 40% calorie deficit	1		1
Total	74		
**How many meal replacement products do you typically prescribe as a daily intake when the goal is rapid weight reduction?**			
	*n*		%
≤3	76		54
Prescribes according to the individual’s height and/or weight *	47		33
4 to 5	19		13
Total	142		
* Please expand on how you structure your prescription of meal replacement products according to the individual’s height and/or weight *(open text)*			
	*n*		%
I prescribed the amount of MRPs according to the individual protein requirements	18		41
I prescribed the amount of MRPs according to the individual protein requirements at an adjusted ideal body weight	7		16
My prescription is based on the severity of obesity	5		11
They are prescribed according to a calculated estimated energy requirement and a predetermined energy deficit to induce weight loss	5		11
I follow commercial product instructions	3		7
They are prescribed according to individual needs or desires	2		5
I prescribed the amount of MRPs according to the health status of the individual	1		2
I give more when needed to satisfy hunger	1		2
I give extra when the person is tall or active	1		2
I prescribed the amount of MRPs according to the amount of fat-free mass	1		2
Total	44		
**Do you ever allow additional items in a meal replacement diet as part of your prescription for weight loss?**			
	*n*		%
Non-starchy vegetables such as spinach, broccoli and tomato, and salad or vegetable soup	146		67
Food-based protein (e.g., meat, fish, eggs, chicken, pork or tofu)	95		44
Fibre	94		43
Diet jelly	86		40
Multivitamin	82		38
Diet soft drinks	78		36
Oil or fat (e.g., butter)	72		33
Diet chewing gum	61		28
Broth	45		21
Omega-3 fatty acids	32		15
Whey or casein protein supplements	27		12
Soy, pea, hemp protein supplements	7		3
Electrolytes	6		3
No additional items are added	5		2
Medium chain triglycerides	2		1

Abbreviations: *n*, number; %, percentage; MRPs, meal replacement products; *, question is related to the previous answer; (*open text*), open text question.

**Table 3 behavsci-10-00136-t003:** Perceptions of healthcare professionals around compliance with, durability of results from, and safety of meal replacement diets.

**Have you ever experienced patient non-compliance with diets involving meal replacement products?**			
	*n*	Yes	%
Total	87	72	83
*** If you have chosen “Yes” what reasons do you believe contribute to patient non-compliance? (*open text*)**			
	*n*		%
Boredom and Emotional Eating	12		19
Taste and Texture of Shakes Deter People	10		16
It is Difficult to Change Behaviour	9		14
The Diet is too Restrictive	9		14
Hunger Gets in the Way	7		11
The Ongoing Cost of Purchasing Products	7		11
Lack of Motivation	5		8
Meal Replacement Diets are not Compatible with Social Occasions	4		6
Total	63		
**What do you believe is the main reason why meal replacement diets result in weight loss? (*open text*)**			
	*n*		%
Decreased Energy Intake	52		61
The Suppression of Appetite from Diet Induced Ketosis	15		18
Structured Program that is Easy to Follow Resulting in Fewer Opportunities to Eat	14		17
I Do Not Know	4		5
Total	85		
What is your experience regarding the long-term outcome of weight loss and weight maintenance when it is achieved with a meal replacement diet? *(open text)*			
	*n*		%
Not Durable Long Term *	99		49
Conditional Durability	46		23
No Long-Term Experience with Meal Replacement Diets	32		16
Durable Short and Long Term	25		12
Total	202		
***The reasons provided for the perceived poor long-term weight loss outcomes of meal replacement diets *(open text)***			
	*n*		%
Meal Replacement Diets did not Encourage Permanent Behaviour Change	25		53
It Is a Highly Restrictive Diet and thus Unsustainable	12		26
All Restrictive Weight-Loss Diets do not Work and Weight Regain in Inevitable	6		13
Lack of Motivation	2		4
Lack of Support	1		2
The Individuals’ Characteristics	1		2
Total	47		
**What are your perceptions about the safety of meal replacement programs as a weight-loss tool? *(open text)***			
	*n*		%
Safe	72		37
Safe with Medical Supervision	60		31
Conditional Safety	41		21
Not Safe	23		12
Total	196		

Abbreviations: *n*, number; %, percentage; MRPs, meal replacement products; *, question is related to the previous answer; (*open text*), open text question.

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
