# Peer review of "Attitudes and Approaches to Use of Meal Replacement Products among Healthcare Professionals in Management of Excess Weight"

_behavsci, 2020, doi:10.3390/bs10090136_

Round 1

Reviewer 1 Report

Dear Authors,

The topic is very interesting and timely because of the increasing number of people with excess weight and an urgent need for more effective fight against it worldwide.

The suggestions and comments below are intended to strengthen the manuscript.

Introduction:

Some information whether there are obesity treatment standards/recommendations in Australia, which include recommendations for the use of MRPs, would be recommended.

Minor comments:

lines 73-74: Missing space or hyphen between "24" and "months"?

line 82: Missing dot.

Methods:

As I understood the survey was carried out with the CAWI method and partially with PAPI. What was the proportion of answers?

Results:

Table 1: Please explain more what you meant by: allied health, medical, specialist medical ect. The table layout should be corrected to fit the journal requirements.

What could be the reason of large differences in the number of respondents from individual states?

I do not fully understand the numbers for n in table 1. If 303 respondents started the survey and 197 completed it (line 147), why, for example, the number of all professional occupations is 269? Please explain.

Table 2 is difficult to follow. Maybe it is worth dividing this table into several tables?

Discussion:

Discussion is a strong point of the manuscript. However, after reading the manuscript, the question arises: who in Australia treats obese people in terms of diet? Is this scope of competence not only provided for a dietitian and MD specialist? If so, is it not pointless to ask nurses or other health care professionals? I would appreciate more information on this.

Author Response

  1. Introduction: Some information whether there are obesity treatment standards/recommendations in Australia, which include recommendations for the use of MRPs, would be recommended.

Our response: We have now included information pertaining to the Australian National Health and Medical Council Clinical Practice Guidelines for the Management of Overweight and Obesity in Adults and the recommended use of meal replacement products in the second last paragraph of the Introduction section.  

  1. Lines 73-74: Missing space or hyphen between "24" and "months"?

Our response: The manuscript has now been amended accordingly.

  1. Line 82: Missing dot.

Our response: The manuscript has now been amended.

  1. Methods: As I understood the survey was carried out with the CAWI method and partially with PAPI. What was the proportion of answers? 

Our response: The predominant method of data collection was via the computer-assisted web interviewing method, with 10% of survey responders captured via pen and paper interviewing. This additional information has now been added to the text, in the first paragraph of the Results section.

  1. Results: Table 1: Please explain more what you meant by: allied health, medical, specialist medical

Our response: All healthcare professionals who responded to our survey were included, thus our categorization of respondents by profession was performed retrospectively by grouping professions by boarder categories. The listed categories are outlined below.  

  • Dietitians and nutritionists
  • Allied health professionals (besides dietitians): physiotherapists, exercise physiologists and exercise scientists, psychologists and counsellors, sonographers, diabetes educators
  • Medical: general practice physicians and nurses
  • Specialist medical: endocrinologists, gynecologists, gastroenterologists, bariatric endoscopists, pediatricians
  • Other healthcare professionals: researchers, pathology laboratory technicians, lymphoedema therapists, public health practitioners

The manuscript has now been revised to include this information in the marked-up text in the Results section.

  1. The table layout should be corrected to fit the journal requirements.

Our response: Tables 1 and 2, have now been revised and divided into 3 separate tables to meet the journal requirements and improve readability.

  1. What could be the reason of large differences in the number of respondents from individual states?

Our response: We were unable to determine why there was a large difference in the number of respondents from individual states.

  1. I do not fully understand the numbers for n in table 1. If 303 respondents started the survey and 197 completed it (line 147), why, for example, the number of all professional occupations is 269? Please explain.

Our response: The question regarding health professional occupation was the first question in the survey, resulting in more people answering this question than subsequent questions. Questions could be skipped if desired and as a result not all of the 303 respondents answered every question. We acknowledge that the phrase “303 participants completed the survey” is thus misleading. This phrase has now been revised to state that “303 participants began the survey”. There were also 45 health professionals that listed more than one professional occupation (e.g., a dietitian who is also dual trained as a diabetes educator), elevating the figures.

  1. Table 2 is difficult to follow. Maybe it is worth dividing this table into several tables?

Our response: Thank you for this suggestion. Table 2 has now been divided into two separate tables to improve readability. The titles of the tables are as follows. “Table 2. Healthcare professionals’ experiences and use of meal replacement products in daily practice” and “Table 3. Perceptions of healthcare professionals around compliance with, durability of results from, and safety of meal replacement diets.”

  1. Discussion is a strong point of the manuscript. However, after reading the manuscript, the question arises: who in Australia treats obese people in terms of diet? Is this scope of competence not only provided for a dietitian and MD specialist? If so, is it not pointless to ask nurses or other health care professionals? I would appreciate more information on this.

Our response: Thank you for this comment. It is common in Australian clinical practice for multidisciplinary teams to work in the management of obesity and co-morbidities. Whilst dietitians and nutritionists are often seen to be gatekeepers of meal replacement diet prescription, other healthcare professionals involved in multi-disciplinary care do prescribe meal replacement diets and have experience with patient compliance, behaviour and satisfaction on such diets. This patient feedback relayed through other healthcare professionals, outside of the dietetic practice, is valuable and provides a more robust understanding of use and prescription barriers.

Reviewer 2 Report

Thank you for the article “Attitudes and approaches to the use of meal replacement products among healthcare professionals in the management of excess weight” There are significant areas for improvement.  I hope you find the following comments helpful:

Please ensure abbreviations are clear with capitals for first use e.g. HealthCare Professionals (HCPs)/ Meal Replacement Products (MRPs)

The Keywords include qualitative but this is the first mention of this as a methodology, the title gives no indication of a qualitative study and the abstract lacks any reference to qualitative methods

The introduction suggests that there has been limited research on Meal Replacement Products, it would be recommended to ensure that the introduction is as up to date as possible and cite some more contemporary evidence on this topic, for example: Astbury NM, Piernas C, Hartmann-Boyce J, Lapworth S, Aveyard P, Jebb SA. A systematic review and meta-analysis of the effectiveness of meal replacements for weight loss. Obes Rev. 2019;20(4):569-587. doi:10.1111/obr.12816

“Despite the redevelopment of MRPs, the adverse historical events seemed to shape the perception of such diets among healthcare professionals (HCPs) into the 21st century (7, 15). In the past decade, three separate surveys show that MRPs and VLEDs remain underused among HCPs. Only 3.2% of dietitians were found to prescribe VLEDs as a weight loss therapy (15). Similarly, just 2% of surveyed British physicians reported they would advise patients to use a VLED (16) and among approximately 1200 surveyed physicians in the USA, meal replacement diets were the least used weight loss therapy (17).”

For the introduction and the discussion: It would be worthy to acknowledge that meal replacements are not typically recommended in obesity/weight loss clinical guidelines, and therefore HCPs who follow such clinical guidelines are unlikely to prescribe/ recommend meal replacement products hence this will impact their behaviour and attitudes. However recent evidence suggests these are effective (See reference above) and therefore could be something to consider.

It is noteworthy that refs 15, 16 , 17 the focus of these studies was obesity management and comparison to best practice guidelines (15), GP management of obesity (16), and Physicians attitudes towards obesity (generally) (17) these studies have not focused specially on meal replacements so it would be helpful to clarify this, and include articles that have looked into this

it would be helpful to clarify the Australian clinical recommendations and evidence base around meal replacements for weight loss given this is where the study has been conducted. 

Might be helpful to put in context the use of meal replacements for weight loss and cite other relevant studies on other professionals such as personal trainers who may recommend use.

Again, put in context the value of meal replacements, either in introduction or as part of a discussion point. …Whist evidence suggests that meal replacements may help weight loss, there is also evidence to show that 50%+ regain this weight. e.g. Ames, GE, Patel, RH, McMullen, JS. Improving maintenance of lost weight following a commercial liquid meal replacement program: a preliminary study. Eat Behav. 2014;15:95-98. doi:10.1016/j.eatbeh.2013.10.022

I think it’s a bold statement to suggest you are exploring the “attitudes and prescription patterns” of meal replacement. You may be exploring perceptions of meal replacement and experiences of prescribing Meal replacements, but I don’t agree you are evaluating prescription patterns.,

Please avoid the term subjects and replace with participants

In terms of recruitment strategy, please clarify who, and provide approximate sample size for each approach. E.g. “professional medical associations”, such as which? Who have what membership size/mail list reach? “private businesses with professional membership” what defines a professional membership please provide details.

Please clarify your definition of working in weight management, and provide a list of the range of professionals you contacted. By definition HealthCare Professionals, if targeting non-health care settings such as commercial business then perhaps the participants were not (registered?) HCPs but people working in the health care industry (such as personal trainers???)

“A sample size was not calculated.” ?? is this referring to a power calculation for appropriate sample size? Again clarify.

“Data was monitored and analysed via preliminary coding in the qualitative responses on a weekly basis.”- what does this mean?

It is unclear what methods were actually applied. There are references to a survey study, a mixed methods study, a qualitative study. Later in data analysis section further details are provided but this is very limited.

In terms of the development of the survey what was the cronbachs alpha the reliability/ validity of the questions asked. In terms of the power and effect size please state.

The survey took an average of 6 minutes to complete and answer 25 questions. How valid do you think the ‘qualitative’ data was?

In terms of completing Thematic analysis please provide further information in terms of actual process of analysis, and as you refer to saturation how this was managed?

For the qualitative data who were the researchers, what was their experience in qualitative methodology/ topic, and how did this influence researcher bias. How was researcher bias mitigated.

The process of TA was “inductive” but the following “When a response did not fit the codes” implies a deductive approach was applied.

However in the actual results section.

There are no inferential statistics, percentages are offered followed by some descriptive commentary. The results therefore lack scientific analysis and the qualitative data presented is not TA.

Unable to read table 2

There are some bold claims made in the discussion based on very limited analysis. There are many limitations of this study and these should be systematically acknowledged within he discussion.

Author Response

  1. Please ensure abbreviations are clear with capitals for first use e.g. HealthCare Professionals (HCPs)/ Meal Replacement Products (MRPs).

Our response: The manuscript has now been amended to reflect this recommendation, specifically in the first line and 3rd paragraph of the Introduction.

  1. The Keywords include qualitative but this is the first mention of this as a methodology, the title gives no indication of a qualitative study and the abstract lacks any reference to qualitative methods

Our response: We have now added text to both the Abstract and the last paragraph of the Introduction stating that the study included qualitative analysis.

  1. The introduction suggests that there has been limited research on Meal Replacement Products, it would be recommended to ensure that the introduction is as up to date as possible and cite some more contemporary evidence on this topic, for example: Astbury NM, Piernas C, Hartmann-Boyce J, Lapworth S, Aveyard P, Jebb SA. A systematic review and meta-analysis of the effectiveness of meal replacements for weight loss. Obes Rev. 2019;20(4):569-587. doi:10.1111/obr.12816

Our response: Thank you, this reference is relevant to our paper and we have added the stated reference in the Introduction, as seen in the marked-up text.

  1. For the introduction and the discussion: It would be worthy to acknowledge that meal replacements are not typically recommended in obesity/weight loss clinical guidelines, and therefore HCPs who follow such clinical guidelines are unlikely to prescribe/ recommend meal replacement products hence this will impact their behaviour and attitudes. However recent evidence suggests these are effective (See reference above) and therefore could be something to consider.

Our response: Thank you for this recommendation. We agree, and indeed, it was one of the reasons for conducting the investigation. We have now added marked-up text to the Introduction section of the manuscript around the conflicting clinical practise guidelines around the world.

  1. It is noteworthy that refs 15, 16 , 17 the focus of these studies was obesity management and comparison to best practice guidelines (15), GP management of obesity (16), and Physicians attitudes towards obesity (generally) (17) these studies have not focused specially on meal replacements so it would be helpful to clarify this, and include articles that have looked into this.

Our response: Thank you, we have included clarification around the more general nature of these references. To our knowledge however, there are no other studies that have investigated physicians’ attitudes specifically to the use of meal replacement diets. Our study aimed to address this gap in the literature.

  1. It would be helpful to clarify the Australian clinical recommendations and evidence base around meal replacements for weight loss given this is where the study has been conducted. Might be helpful to put in context the use of meal replacements for weight loss and cite other relevant studies on other professionals such as personal trainers who may recommend use. Again, put in context the value of meal replacements, either in the introduction or as part of a discussion point.

Our response: Thank you for providing this advice, we have now added text in the Introduction to reflect this.  

  1. I think it’s a bold statement to suggest you are exploring the “attitudes and prescription patterns” of meal replacement. You may be exploring perceptions of meal replacement and experiences of prescribing Meal replacements, but I don’t agree you are evaluating prescription patterns.

Our response: We have now couched this statement in section 2.2, such that it reads “It consisted of both quantitative and qualitative questions to capture perceptions and experiences of HCPs in prescribing MRPs.”

  1. Please avoid the term subjects and replace with participants

 Our response: Thank you, the manuscript has now been amended accordingly.

  1. In terms of recruitment strategy, please clarify who, and provide an approximate sample size for each approach. E.g. “professional medical associations”, such as which? Who have what membership size/mail list reach? “private businesses with professional membership” what defines a professional membership please provide details.

Our response: As a result of the nature of such organizations and limitations in obtaining private business knowledge, we do not know the size of the membership base or mail list reach. The manuscript has now been amended to reflect this.

  1. Please clarify your definition of working in weight management, and provide a list of the range of professionals you contacted. By definition HealthCare Professionals, if targeting non-healthcare settings such as commercial business then perhaps the participants were not (registered?) HCPs but people working in the health care industry (such as personal trainers???)

Our response: HCPs were defined as university-trained medical professionals. We defined HCPs who worked in weight management as those who practised in the private or public healthcare system, commercial business, research that provided a medical service of any type in conjunction with weight loss advice or a weight loss service as a stand-alone service.

Private business with professional memberships included professional education providers such as Dietitian Connection and MegaBite Nutrition. Professional medical associations included Dietitians Association of Australia Obesity Interest Group, Exercise Sports Science of Australia, The Australian and Primary Care Nurses Association, and to all Australian state Primary Health Networks. The manuscript has now been amended to include this additional information, in the Methods section.

  1. “A sample size was not calculated.” ?? is this referring to a power calculation for appropriate sample size? Again clarify.

Our response: This statement has been deleted and replaced with “a response rate could not be calculated” and further expanded to clarify in the methods section.

  1. “Data was monitored and analysed via preliminary coding in the qualitative responses on a weekly basis.”- what does this mean?

Our response: Thank you for identifying this ambiguity in our reported research methods. We have now added additional text to clarify the data monitoring to the methods section.

  1. It is unclear what methods were actually applied. There are references to a survey study, a mixed-methods study, a qualitative study. Later in data analysis section, further details are provided but this is very limited.

Our response: The initial analysis included familiarisation with the data in open-text responses in the survey by reviewing each response, then reviewing a second time to create a list of descriptive categories for each question. In a third review, a code was then applied to each unique descriptive category and revised, and we then collapsed descriptive categories into broader theme-based categories. Responses were then reviewed a fourth time to determine the fit of the responses to each code. Each collapsed category was then numerically quantified by the number of respondents and frequency in which the category was identified providing descriptive statistics to open-text survey responses. Further analysis included comparing theme-base categories across the entire survey questionnaire to further collapse categories into key overarching concepts related to barriers that may prevent MRP use. The methods section of the manuscript has now been revised to provide clarity around our qualitative analysis methodology.

  1. In terms of the development of the survey what was the cronbachs alpha the reliability/ validity of the questions asked. In terms of the power and effect size please state.

Our response: We did not calculate a Cronach’s alpha score, because it is a measure more commonly used to test the reliability of psychometric surveys that use multiple Likert-scale questions that are quantitative in nature. Our questionnaire used a combination of dichotomous, multiple-choice and open-text questions; thus, this calculation would not be applicable. To improve the reliability of our survey instrument we used a peer-review process, using a test-retest method, which included a peer review of the proposed questionnaire whereby 10 experienced HCPs working in weight management provided feedback about the readability, validity and composition of questions. The questionnaire was revised with the feedback included and returned to the group of HCPs for final review. This has been outlined in section 2.2 “Measures”.

  1. The survey took an average of 6 minutes to complete and answer 25 questions. How valid do you think the ‘qualitative’ data was?

Our response: The measurement of 6 minutes was an average calculation, with time taken to complete the survey ranging from approximately 1 to 20 minutes. We have now added a time range to section 2.2, which is subtitled “Measures”. There were 12 open-text qualitative based questions and responses ranged in length from 1 to 4 sentences in length. We have now added text to the limitations section of our Discussion stating that the qualitative aspect of this study is limited because of the nature of the open-text questions and the brief answers that were provided.

  1. In terms of completing Thematic analysis please provide further information in terms of actual process of analysis, and as you refer to saturation how this was managed?

Our response: Thank you for identifying this gap in our reported methodology. We have removed the words “Thematic analysis”, we have also added more information to clarify our research methodology. The open-text survey responses were monitored and analysed for themes and repetition thereof on a weekly basis. Although data saturation occurred when the 249th survey participant completed the survey, the survey was not discontinued at that point due to the collection of quantitative data points contained in the survey.

  1. For the qualitative data who were the researchers, what was their experience in qualitative methodology/ topic, and how did this influence researcher bias. How was researcher bias mitigated?

Our response: Thank you for identifying this gap in our reported methodology. We have now added more information in section 2.3 “Data analysis” to clarify our research experience and how we attempted to mitigate researcher bias through a peer review, supervision and consultation process.

  1. The process of TA was “inductive” but the following “When a response did not fit the codes” implies a deductive approach was applied.

Our response: Thank you, we acknowledge that this statement is confusing. We have now added more information to clarify our research methods in section 2.3 “Data analysis”. It now reads “An inductive approach was applied to create descriptive coding lists for each open-text response. When the inferred meaning of the open-text response could not be adequately determined, a second investigator, J.F., was consulted and after review, a code was applied or a new code was created.”

  1. There are no inferential statistics, percentages are offered followed by some descriptive commentary. The results therefore lack scientific analysis and the qualitative data presented is not thematic analysis.

Our response: In section 3.2.3 inferential statistics were used. A Relative Risk calculation and a hierarchical multiple regression analysis was conducted to determine if there was a relationship between meal replacement use and HCP category, place of work, and whether or not they had formal training. This analysis included data points from multiple questions from both our demographic information and area of clinical practice questions and prescription patterns questions. We felt that there were no other interesting or meaningful areas to conduct any further inferential statistic analysis.

In section 2.3, sub-titled “Data analysis”, we have now removed the word “thematic” and replaced it with the word “secondary”. Additional text has been added to clarify our qualitative analysis processes.

  1. Unable to read table 2

Our response: Table 2 has now been revised and split into two separate tables (2 and 3) to improve readability.

  1. There are some bold claims made in the discussion based on very limited analysis. There are many limitations of this study and these should be systematically acknowledged within he discussion.

Our response: Thank you for this recommendation. We have listed further limitations of this study in the Discussion section as seen in the marked-up text. 

Reviewer 3 Report

Major concern:

Personal characteristics may affect the attitude and approaches to the use of MRP. For example, the year of employment duration, educational background (knowledge of nutrition as an example), sex and age, et al., these information were important, however, the authors did not provide them.

Minor concern:

  1. Line 20 “The survey was completed by 303 HCPs working in weight management across Australia”, but Line 147 said “… 197 completed…”. Please making sure line 20 provided the right information.
  2. Did all participants finish the investigation online? How you make sure they finished the investigation by themselves?
  3. Line 98: “Responders were encouraged to share the survey with colleagues working in weight management resulting in recruitment by snowballing” What the proportion of the participants who coming from the same work place? Did this factor bias the result?
  4. Line 163: “The full results of the survey are provided in Table 2”. I noted your survey contained 25 questions. Did the table 2 present all results?

Author Response

  1. Personal characteristics may affect the attitude and approaches to the use of MRP. For example, the year of employment duration, educational background (knowledge of nutrition as an example), sex and age, et al., this information were important, however, the authors did not provide them.

Our response: Thank you for this observation. Data on the personal characteristics of research participants, such as duration of employment, sex and age, were not collected. We have amended our Discussion to list this a limitation to our investigation. As for educational background, we have appropriately captured this by asking respondents to list their professional occupation and allowing them to list more than one professional occupation.

  1. Line 20 “The survey was completed by 303 HCPs working in weight management across Australia”, but Line 147 said “… 197 completed…”. Please making sure line 20 provided the right information.

Our response: Thank you for this recommendation. We have now been amended this error in the manuscript.

  1. Did all participants finish the investigation online? How do you make sure they finished the investigation by themselves?

Our response: Not all participants finished the survey, as indicated by the variability in the number of responses for each question. We do not know if the survey was initiated and completed by the same person. We have now added this limitation to our Discussion.

  1. Line 98: “Responders were encouraged to share the survey with colleagues working in weight management resulting in recruitment by snowballing” What’s the proportion of the participants who coming from the same work place? Did this factor bias the result?

Our response: As the survey was anonymous, we could not quantify the proportion of participants who responded from the same work place. We have now added this information as a limitation to our study in the Discussion section as seen in the marked-up text.

  1. Line 163: “The full results of the survey are provided in Table 2”. I noted your survey contained 25 questions. Did the table 2 present all results?

Our response: The survey table did not present all the results. As stated in the Methods section, parts of the survey were collapsed into broader themes to avoid repetition of results. We have now amended this statement to clarify this.

Round 2

Reviewer 2 Report

Thank you for systematically addressing my commentary. The manuscript has been improved. 

I think you could have made more of your quantitative analysis and also qualitative analysis could have been more analytical/interpretative.  You could have considered more of a theoretical understanding in the discussion to meal replacement use, although overall the article is ok.